

# Splitting on categorical predictors in random forests

Marvin N. Wright[1] and Inke R. König[2]

[1] Leibniz Institute for Prevention Research and Epidemiology—BIPS, Bremen, Germany
[2] Institut für Medizinische Biometrie und Statistik, Universität zu Lübeck, Universitätsklinikum Schleswig-Holstein, Campus Lübeck, Lübeck, Germany

## ABSTRACT

One reason for the widespread success of random forests (RFs) is their ability to analyze most datasets without preprocessing. For example, in contrast to many other statistical methods and machine learning approaches, no recoding such as dummy coding is required to handle ordinal and nominal predictors. The standard approach for nominal predictors is to consider all $2^{k-1} - 1$ 2-partitions of the $k$ predictor categories. However, this exponential relationship produces a large number of potential splits to be evaluated, increasing computational complexity and restricting the possible number of categories in most implementations. For binary classification and regression, it was shown that ordering the predictor categories in each split leads to exactly the same splits as the standard approach. This reduces computational complexity because only $k - 1$ splits have to be considered for a nominal predictor with $k$ categories. For multiclass classification and survival prediction no ordering method producing equivalent splits exists. We therefore propose to use a heuristic which orders the categories according to the first principal component of the weighted covariance matrix in multiclass classification and by log-rank scores in survival prediction. This ordering of categories can be done either in every split or a priori, that is, just once before growing the forest. With this approach, the nominal predictor can be treated as ordinal in the entire RF procedure, speeding up the computation and avoiding category limits. We compare the proposed methods with the standard approach, dummy coding and simply ignoring the nominal nature of the predictors in several simulation settings and on real data in terms of prediction performance and computational efficiency. We show that ordering the categories a priori is at least as good as the standard approach of considering all 2-partitions in all datasets considered, while being computationally faster. We recommend to use this approach as the default in RFs.

# INTRODUCTION

Random forests (RF; *Breiman, 2001*) are a popular machine learning method, successfully used in many application areas such as economics (*Varian, 2014*), spatial predictions (*Hengl et al., 2017*; *Schratz et al., 2018*) or genomics (*Goldstein, Polley & Briggs, 2011*). In part, their success is due to their ease of use: RF often works well without much fine tuning or preprocessing, and categorical predictors can be included without recoding.

Corresponding author
Marvin N. Wright,
wright@leibniz-bips.de

Ordered categorical (ordinal) predictors such as customer satisfaction or the Likert scale can simply be treated the same way as numerical predictors.

For unordered categorical (nominal) predictors such as geographical regions, colors, medication types or cell types, the standard approach is to consider all 2-partitions of the $k$ predictor levels. Each of these $2^{k-1} - 1$ partitions is tried for splitting, and the best partition is selected. This approach has two major drawbacks: First, for predictors with many categories, the exponential relationship produces a large number of potential splits to evaluate, increasing computational complexity. For example, the predictor *country code* with all 28 EU countries as categories would produce $2^{28-1} - 1 = 1.34 \times 10^8$ possible split points. Second, each category has to be assigned to the left or right child node in every node and thus these partition splits cannot be saved as a single split point, as it is done for ordinal or continuous predictors. A popular option is to use the bit representation of a single integer to save these child node assignments, for example, by setting a bit to 0 for the left child node and to 1 for the right child node. However, this approach limits the possible number of categories to 32 or 64, depending on the implementation.

As a solution for the cases of for binary classification and regression, it was suggested to order the predictor levels in each split. Specifically, it was shown that ordering the categories by the proportion of 1's in a binary outcome and treating these ordered categories as ordinal leads to exactly the same splits in the optimization (*Fisher, 1958*; *Breiman et al., 1984*; *Ripley, 1996*). The same holds for ordering by increasing mean of continuous outcomes (*Breiman et al., 1984*). This approach reduces computational complexity, because only $k - 1$ splits have to be considered for a nominal predictor with $k$ categories.

An extension to ordering predictor levels at every split is to order them only once before growing the forest. With this approach, the split point reduces to a single number, as for ordinal or continuous predictors. In addition, the categories have to be sorted only once, thus speeding up the computation. For multiclass classification and survival prediction, the simplifications of *Breiman et al. (1984)*, *Ripley (1996)* and *Fisher (1958)* do not apply and no fast method which selects exactly the same splits exist. Several methods have been proposed to reduce the computational burden. *Nádas et al. (1991)* and *Mehta, Agrawal & Rissanen (1996)* propose heuristics which iteratively find a good split but the splits have to be saved as node assignments. This leads to limits in the number of levels and to memory problems (see above), and the methods cannot be used to order categories a priori. *Loh & Vanichsetakul (1988)* propose to transform the categorical variable to a dummy matrix and project the columns of this matrix onto their largest discriminant coordinate (CRIMCOORD) (*Gnanadesikan, 2011*). *Coppersmith, Hong & Hosking (1999)* proposes to order the categories according to the first principal component of the weighted covariance matrix. We use the approach of *Coppersmith, Hong & Hosking (1999)* for multiclass classification because it can be applied a priori to the entire dataset and is computationally faster than the approach of *Loh & Vanichsetakul (1988)*. For survival outcomes, we propose to order by the log-rank score, as defined by *Hothorn & Lausen (2003)*.

We compare the two ordering schemes and the proposed methods for multiclass classification and survival prediction with the standard approach in simulations and on real data in terms of prediction performance and computational efficiency. As a benchmark, we also include dummy coding as it is used in generalized linear models. We show that the choice of the splitting algorithm for nominal predictors has a great impact on prediction performance and runtime. Compared to the standard approach, the performance can be improved while simultaneously reducing computational time.

## METHODS

### Random forests

Random forests (*Breiman, 2001*) are a machine learning method based on collections of classification, regression or survival trees. In each split within a tree, the impurity reduction is maximized. This reduction is usually measured by the Gini index (*Breiman et al., 1984*) for classification, by the sum of squares (*Ishwaran, 2015*) for regression and by the log-rank statistic (*Ishwaran et al., 2008*) or Harrell's C-index (*Schmid, Wright & Ziegler, 2016*) for survival outcomes. To create diverse trees, each tree is grown on a subsample or bootstrap sample of the data. A further random element is introduced by randomly drawing splitting candidate variables at each split. The algorithm grows trees to form the ensemble and aggregates the predictions of the single trees to generate an ensemble prediction. The aggregation procedure is specific to the outcome type: For classification a majority vote is used, for regression an arithmetic mean and for survival an ensemble cumulative hazard estimator (*Ishwaran et al., 2008*). For details we refer to the original literature (*Breiman, 2001*) and to recent reviews (*Biau & Scornet, 2016*; *Boulesteix et al., 2012*; *Ziegler & König, 2014*).

The most important parameters of RFs are the number of trees, the number of variables randomly selected as splitting candidates (the so-called *mtry* value) and tree size (*Probst, Wright & Boulesteix, 2018b*). The tree size can be specified by setting a lower bound for the terminal node size or an upper bound for the tree depth. For classification outcomes, no average has to be computed in the terminal nodes and consequently tree growing typically continues until only one class remains in a node. Cross validation can be used to tune these parameters in order to optimize prediction accuracy without overfitting. Model comparisons should use nested resampling for unbiased results (*Bischl et al., 2012*). According to *Probst, Bischl & Boulesteix (2018a)*, *mtry* is the parameter with the highest tunability, that is, is most important to tune.

### Methods to handle nominal predictors

#### Standard approach: consider all 2-partitions

The standard approach to handle nominal predictors in RFs is to consider all 2-partitions of the $k$ predictor categories. Each category can either be assigned to the left or the right child node. The order of the two child nodes does not matter for further splitting and thus the number of possible partitions is calculated by the Stirling partition number $S(k, 2) = 2^{k-1} - 1$. Each of these partitions is tried for splitting, and the best partition is selected, see Fig. 1A for a simple example. We refer to this method as "Partition."
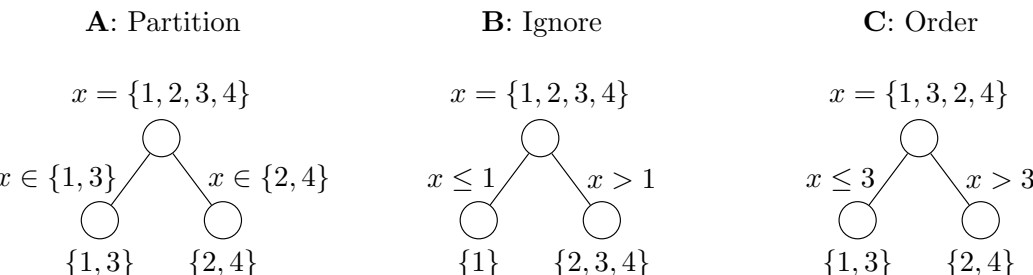

**Figure 1 Illustration of nominal predictor methods.** The task in this example is to separate even and odd digits in a single split. The Partition method (A) tries all seven possible partitions and selects the correct one. In Ignore (B), all three possible linear splits are tried but the correct partition is impossible. The Order method (C) orders the categories before splitting and is able to find the correct split with three trials.

A memory efficient way to save these splits is to save node assignment as integer bit representation. For each category one bit is set to 0 or 1, corresponding to an assignment to the left or right child node, respectively. Because there are $2^{k-1} - 1$ possible 2-partitions, only 64 categories can be assigned with a 64 bit integer. In some implementations such as `randomForest` (*Liaw & Wiener, 2018*) and `ranger` (*Wright & Ziegler, 2017*), a double is used to save the split points, inducing a limit of 53 levels[1]. Alternatively, all levels assigned to one child have to be saved in a split, removing the level limit but requiring more memory.

[1] Up to randomForest version 4.6–8 there was a limit of 32 categories.

### Ordering categories

As explained above, it was shown for continuous outcomes that ordering the predictor levels in each split by increasing mean of the outcome and treating the ordered categories as ordinal leads to exactly the same splits in the optimization as the partitioning approach (*Breiman et al., 1984*; *Fisher, 1958*). For binary outcomes, splitting by the Gini impurity is equivalent to coding the outcome as 0 and 1 and applying the weighted variance splitting used for regression outcomes (*Ishwaran, 2015*). Consequently, the same equivalence of splits as for continuous outcomes holds (*Ripley, 1996*; *Breiman et al., 1984*).

For multiclass classification, no sorting algorithm leading to equivalent splits is known and we propose to use the approximation of *Coppersmith, Hong & Hosking (1999)*. To sort the nominal predictor $x$ with the categories $1, \ldots, k$ by the multiclass outcome $y$ with the classes $1, \ldots, c$, the method works as follows:

Step 1 Compute the $k \times c$ contingency table $N$ between the predictor $x$ and the outcome $y$, where $k$ is the number of categories of $x$ and $c$ the number of classes of $y$.

Step 2 Convert the contingency table $N$ to the class probability matrix $P$, where each row $p^\alpha$, $\alpha \in 1, \ldots, k$ consists of the elements

$$p_\gamma^\alpha = \frac{1}{\sum_{i=1}^n \mathbb{1}_{x_i=\alpha}} \sum_{i=1}^n \mathbb{1}_{x_i=\alpha \wedge y_i=\gamma}, \qquad \alpha \in 1, \ldots, k, \quad \gamma \in 1, \ldots, c,$$

representing the relative class frequencies of predictor category $\alpha$.

Step 3 Compute the weighted covariance matrix

$$\mathbf{\Sigma} = \frac{1}{n-1} \sum_{\alpha \in A} n_\alpha (\boldsymbol{p}^\alpha - \bar{\boldsymbol{p}})(\boldsymbol{p}^\alpha - \bar{\boldsymbol{p}})^T,$$

where $n_\alpha = \sum_{i=1}^n \mathbb{1}_{x_i = \alpha}$ is the number of samples with category $\alpha$, $n$ is the total number of samples and $\bar{\boldsymbol{p}}$ is the vector of mean class probabilities per outcome class.

Step 4 Compute the first principal component $\boldsymbol{v}$ of $\mathbf{\Sigma}$ and the principal component scores $S_\alpha$ of the predictor categories by $S_\alpha = \boldsymbol{v} \cdot \boldsymbol{p}^\alpha$.

Step 5 Sort the categories by their principal component scores $S_\alpha$.

As in the case of multiclass classification, no lossless alternative to partitioning is known for survival outcomes. To order categories by their average survival time, censoring has to be taken into account. A simple approach would be to use the median survival, as estimated by the Kaplan–Meier method. However, in case of high censoring proportions, more than half of the observations may still be alive at the end of the study, so that the median survival will not be available. As an alternative, we propose to sort by the mean of log-rank scores (*Hothorn & Lausen, 2003*) of each category. The log-rank score for an observation $i$ is defined as

$$a_i(\mathbf{Z}, \boldsymbol{\delta}) = \delta_i - \sum_{j=1}^{\gamma_i(\mathbf{Z})} \frac{\delta_j}{(N - \gamma_i(\mathbf{Z}) + 1)},$$

where $\mathbf{Z}$ and $\boldsymbol{\delta}$ are the vectors of survival time and censoring indicators, respectively, and $\gamma_i(\mathbf{Z})$ is the number of observations which experienced an event or were censored before or at time $Z_i$ (*Hothorn & Lausen, 2003*).

The equivalence of the partitioning and ordering approach for continuous and binary outcomes holds if the categories are re-ordered in each split in the tree. An alternative is to order the categories a priori, that is, once on the entire dataset before the analysis. This approach has several advantages: First, the categories have to be sorted only once, resulting in faster computation. Second, this approach does not suffer from the category limit problem—the level order just needs to be saved once for each nominal predictor. Third, since the categories are treated as ordinal in the splitting procedure, categories not present in the training data in a node can still be assigned to a child node and thus the "absent levels problem" (*Au, 2017*) is avoided. Finally, the approach can be easily used with any RF implementation because no changes in the splitting algorithm are required.

In the following, we denote ordering in each split or a priori as "Order (split)" and "Order (once)," respectively. We implemented both ordering schemes in the `simpleRF` package (*Wright, 2018*). The package uses only plain R to allow easy modification of the RF splitting algorithm which is required for the Order (split) method. We also implemented the Order (once) method in the runtime-optimized `ranger` package (*Wright & Ziegler, 2017*).

### Other alternatives

Another alternative is to use dummy or one-hot encoding. In one-hot encoding, a nominal predictor with $k$ categories is replaced by $k$ binary predictors, each corresponding to

**Table 1 Simulation scenarios.**

| Name | Categories | Predictor type |
|------|-----------|----------------|
| Digit | 10 | Digits between 0 and 9, separate odd and even digits |
| SNP | 3 | Single nucleotide polymorphisms with heterozygous effect |
| Group | 10 | Two groups of observations with different predictor effects |

Note:
Three predictor types were combined with four types of outcome: Regression, binary classification, multiclass classification, and survival (described in text). Each scenario was run with $p = 4$ predictors and sample sizes of $n \in \{50,100,200\}$.

one category. For each observation, one of these $k$ binary predictors is 1, all others are set to 0. The same information contained in the $k$ binary predictors can be saved in $k - 1$ binary predictors by setting all converted predictors to 0 for one category (the reference category). Usually, the former is denoted one-hot encoding and the latter dummy coding. The advantages of these approaches are their simplicity and that they can be used with any algorithm able to handle numeric data. However, the size of the dataset increases substantially, in particular for many categorical variables or for many categories. We denote this method as "Dummy."

Finally, the simplest approach is to ignore the nominal nature of the predictors. They are assumed to be ordinal in the order they are encoded or as they appear in the data. For example in R, factor variables are sorted alphabetically by default. We refer to this method as "Ignore." An example is shown in Fig. 1B.

### Methods available in other packages

The package `randomForest` (*Liaw & Wiener, 2018*) orders the levels of a nominal predictor in every split for regression and binary classification but considers all 2-partitions for multiclass classification. The `randomForestSRC` package (*Ishwaran & Kogalur, 2017*) considers all 2-partitions for all outcome types. In `h2o` (*H2O.ai, 2017*), several options to handle nominal predictors are available, including one-hot encoding and ordering a priori. However, no proper ordering method is implemented for multiclass classification. By default, the categories are grouped to fewer categories (in lexical ordering) in each split until the number of categories is below a threshold, in which case all 2-partitions are considered. Many other implementations, including `xgboost` (*Chen et al., 2018*) and the Python implementation `scikit-learn` (*Pedregosa et al., 2011*), currently accept only numeric data. However, for `scikit-learn` it has been discussed whether support for nominal data should be added (https://github.com/scikit-learn/scikitlearn/pull/4899).

### Simulation studies

We performed simulation studies to compare the methods to handle nominal predictors regarding prediction performance and computational efficiency. Three types of nominal predictors were combined with four types of outcome, resulting in twelve simulation scenarios (Table 1).

The first type of nominal predictors were digits. For each of the $p$ predictors, $n$ digits between 0 and 9 were simulated. For regression, the outcome was the difference between the number of odd digits and the number of even digits, for binary classification

the outcome coded whether there were more odd than even digits. For multiclass classification, one outcome class was used for each combination of the number of odd and even digits, corresponding to 16 classes. For survival outcomes, an exponentially distributed random variable was used for the survival time, where the parameter $\lambda$ was set to the sum of the difference between the number of odd digits and the number of even digits (as in regression). A fraction of 30% of the training observations were randomly selected to be censored. The time until censoring for these observations was drawn from a uniform distribution between 0 and the true survival time. The validation data was not censored.

The second type of nominal predictors were single nucleotide polymorphisms (SNPs), that is, variations in single nucleotides on the genome. SNPs are usually coded with the number of minor alleles, which can be 0, 1, or 2. We simulated a heterozygous effect, that is, an effect if exactly one minor allele is present. For regression, we simulated the outcome with a simple linear model, for binary classification with a logit model. For multiclass classification, one outcome class was used for each combination of heterozygous effects, corresponding to 16 classes. For survival outcomes, the same procedure as in the Digit data was used with the parameter $\lambda$ set to the linear predictor used in the regression case. We used $p = 4$ predictors per dataset, a minor allele frequency of 25% and an effect size of 3.

The third type of nominal predictors was specifically designed without a latent ordering to make a priori ordering difficult. Each of the $p$ predictors was simulated with 10 equally frequent categories. The categories were simulated to have different linear effects in two groups of observations. All $n$ observations were randomly assigned to one of two groups and in both groups each category was assigned a random effect size between 1 and 10. For regression, the outcome was the sum of the predictor effects. For binary classification the outcome coded whether this sum was larger than the median of all observations. For multiclass classification, each predictor was assigned to be *high* if the effect size was 5 or higher or *low* for values below 5. Each combination of *high* and *low* values corresponded to one outcome class, with a total of 16 classes. For survival outcomes, the same procedure as for the other predictor types was used, where the parameter $\lambda$ was set to the sum of the predictor effects.

We applied all five methods to handle nominal predictors on each simulation scenario. For each dataset, we simulated $n$ training observations and $n$ validation observations. We used nested resampling for performance evaluation: 100 training and validation datasets were simulated and on each training dataset, we tuned the *mtry* value, that is, the number of variables available for splitting in a node, with a five-fold cross validation. Ten values evenly distributed between 1 and the number of predictors were used for the dummy coding and $mtry \in \{1,2,3,4\}$ for all other approaches. The number of trees was set to 50. All other parameters were set to the default values. In each simulation replicate, the tuned model was then applied to the validation data, to evaluate prediction performance. For regression, the mean squared error (MSE) was used to measure prediction performance, for classification the proportion of misclassifications and for survival the integrated Brier score (*Graf et al., 1999*). The simulations were repeated with

**Table 2 Real datasets analyzed in the comparison of prediction performance and runtime.**

| Name | Type | Obs. | Vars. | Cat. |
|------|------|------|-------|------|
| Tic-tac-toe | Binary classification | 958 | 9 | 3 |
| Splice | Multiclass classification (three classes) | 3,190 | 60 | 4–6 |
| MPG | Multiclass classification (seven classes) | 234 | 10 | 3–38 |
| Servo | Regression | 167 | 4 | 4–5 |
| RA SNPs | Binary classification | 500 | 501 | 3–4 |
| AIDS | Survival | 478 | 4 | 2–8 |

Note:
The name, type, number of observations (Obs.), number of predictor variables (Vars.), and the number of predictor categories (Cat.) are shown.

$n \in \{50, 100, 200\}$. In each simulation scenario, the alternative methods were statistically compared to the Partition method with a two-sided paired $t$-test over the simulation replications. Differences with two-sided $p < 0.05$ were considered statistically significant.

To study the effect of censoring in the survival case, we performed an additional simulation study. All simulation settings were unchanged except the censoring proportion was varied between 0 (no censoring) and 0.9 (90% censored observations) in steps of 0.1. Further, we conducted two simulations to understand the impact of the nominal predictor methods on the splitting procedure on multiclass data. In the first simulation, we compared the size of trees grown with the Ignore and the Order (once) methods, measured as the number of nodes in a tree. In the second simulation, we investigated the "absent levels problem" (*Au, 2017*), that is, that after several splits in a tree some categories are not available anymore and with most splitting methods no meaningful split is possible. For this, we calculated the number of available categories for a given tree size.

In all simulations, we measured runtime on a 64-bit Linux platform with two 8-core Intel Xeon E5649 2.53GHz CPUs. All simulations were performed with the `simpleRF` package (*Wright, 2018*).

## Real data analysis

To compare prediction performance and computational efficiency of the approaches, we analyzed six real datasets. The datasets are summarized in Table 2. The datasets Tic-tac-toe, Splice, MPG, and Servo are standard benchmark datasets from the UCI machine learning repository (*Lichman, 2013*). These datasets were obtained from OpenML (*Vanschoren et al., 2014*).

The Tic-tac-toe dataset contains all possible board configurations at the end of a tic-tac-toe game. The classification task is to predict whether the starting player wins the game. The nine predictor variables each code one position of the board, which can be blank or taken by either player. The objective in the Splice dataset (*Noordewier, Towell & Shavlik, 1991*) is to recognize and classify DNA sequences containing a boundary between exon and intron or vice versa. The outcome variable has the possible classes *ei*, *ie*, or *n*, corresponding to the boundaries between exon and intron, intron and exon or neither. Each predictor variable represents a position in a 60-residue DNA sequence and is coded by the four nucleobases. The MPG dataset contains data on cars from the years 1999

to 2008. The objective is to predict the class of a car, based on the other attributes such as manufacturer, transmission type, or fuel consumption. It includes nominal as well as numerical data. The Servo dataset is obtained from a simulation of a servo system (*Quinlan, 1992*). The outcome is the time required for the system to respond to a step change (rise time), measured as a continuous variable. The four predictor variables are two gain settings and two choices of mechanical linkage. In the RA SNPs dataset, the task is to predict rheumatoid arthritis affection status based on genetic data. The analyzed dataset is part of that used in the Genetic Analysis Workshop 16 (*Amos et al., 2009*). It includes 500 randomly selected observations (250 cases and 250 controls) with genotypes of 500 randomly selected SNPs from the HLA region on chromosome 6, which is known to be associated with rheumatoid arthritis (*Amos et al., 2009*). The AIDS dataset (*Venables & Ripley, 2002*) contains survival data from patients diagnosed with AIDS in Australia before July 1, 1991. Predictors are the patient's sex and age (in years), the state of origin (grouped into four categories) and the transmission category (eight categories). To balance the categories and reduce computational time, we randomly sampled 100 observations from the transmission category *hs* (male homosexual or bisexual contact) and included all observations from other transmission categories, resulting in 478 observations.

We analyzed all datasets with the five methods to handle nominal predictors. We used a 10/5-fold nested cross validation (repeated 50 times) for performance evaluation. In the inner five-fold cross validation, the *mtry* value was tuned (10 values evenly distributed between 1 and the number of predictors or all possible values if less than 10 predictors). In the outer 10-fold cross validation the prediction performance was evaluated with the MSE for regression, with the proportion of misclassifications for classification and with the integrated Brier score for survival. The number of trees was set to 50. All other parameters were set to the default values. On each dataset, the alternative methods were statistically compared to the Partition method with a two-sided corrected paired *t*-test for repeated cross validation (*Nadeau & Bengio, 2003*). Differences with two-sided $p < 0.05$ were considered statistically significant. As in the simulation study, runtime was measured with the `simpleRF` package (*Wright, 2018*) on a 64-bit Linux platform with two 8-core Intel Xeon E5649 2.53GHz CPUs.

## RESULTS

### Simulation studies

The simulation results for a regression outcome and a sample size of $n = 100$ are shown in the first row of Fig. 2, the corresponding runtimes in Table 3 and Fig. S3 and the results of statistical testing in Table S1. On the Digit data, Order (once) performed best, followed by Order (split) and Partition. The Dummy and Ignore methods performed worse, with a larger distance. In terms of runtime, Order (once) and Ignore were fastest, followed by Order (split) and Dummy, while the Partition method was slowest. On the SNP data, the differences in prediction performance were smaller and Dummy performed as well as Order (once). Partition and Dummy were faster on the SNP data, as expected because of the lower number of categories. Here, Order (split) was the slowest

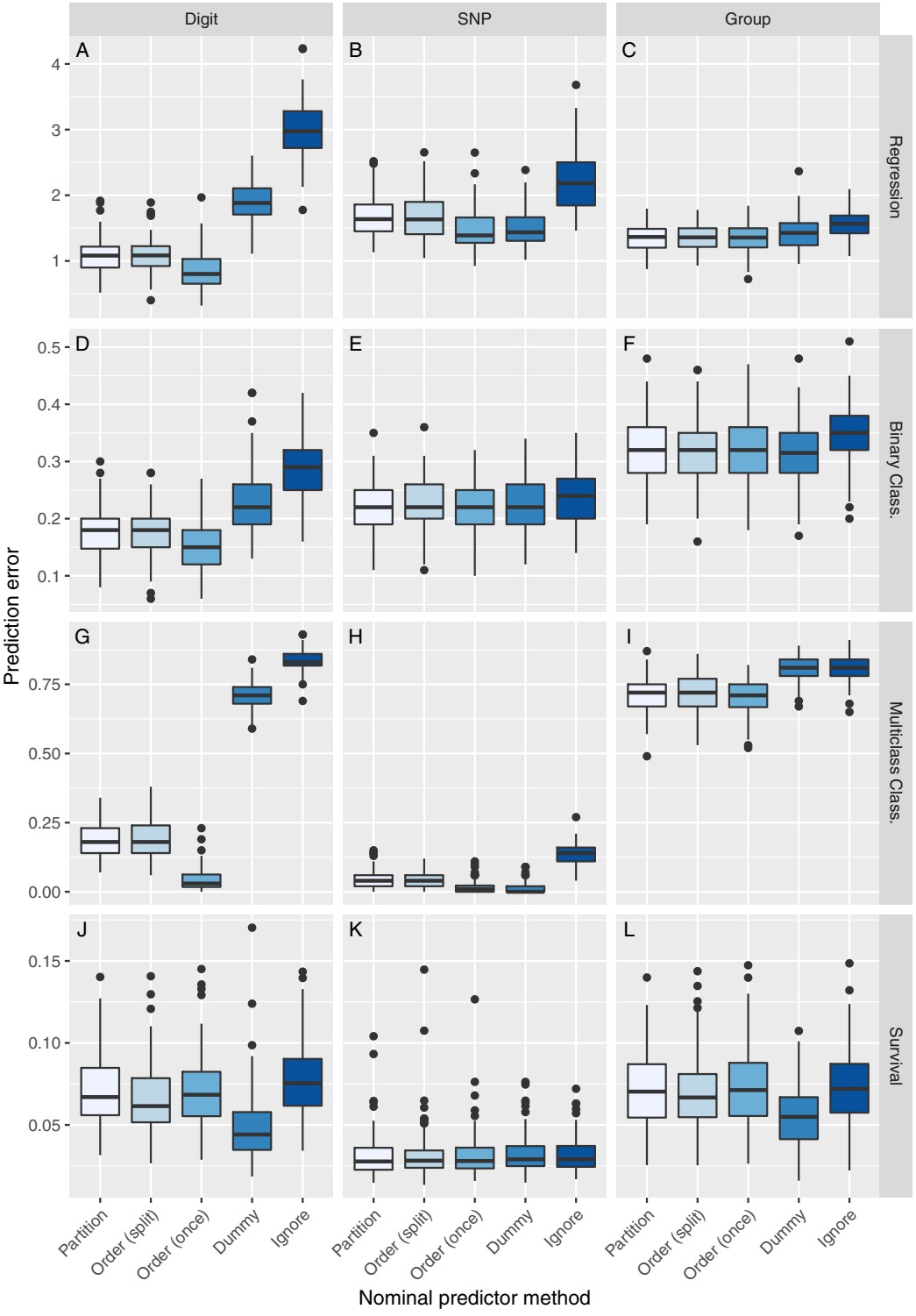

**Figure 2 Simulation results for *n* = 100.** The vertical panels correspond to the predictor type (Digit, SNP, or Group), the horizontal panels to the outcome type (regression, binary classification, multiclass classification, or survival). Each boxplot represents the prediction error for one method to handle nominal predictor variables, applied to one simulation scenario (A–L). The prediction error is measured by the mean squared error for regression, by the proportion of misclassifications for classification and by the integrated Brier score for survival.

**Table 3 Median runtimes of simulations (in seconds).**

| Predictor type | Outcome type | Nominal predictor method | | | | |
|---|---|---|---|---|---|---|
| | | Partition | Order (split) | Order (once) | Dummy | Ignore |
| Digit | Regression | 15.89 | 6.22 | **4.69** | 6.60 | 4.93 |
| | Binary class. | 18.23 | 4.27 | **3.80** | 5.92 | 5.62 |
| | Multiclass class. | 28.47 | 11.09 | **5.77** | 13.19 | 9.84 |
| | Survival | 660.43 | 74.85 | 61.88 | **9.24** | 51.45 |
| SNP | Regression | 4.54 | 6.01 | 4.72 | **4.13** | 4.48 |
| | Binary class. | 3.33 | 4.36 | 3.55 | **2.89** | 3.44 |
| | Multiclass class. | 4.27 | 7.14 | 4.03 | **3.98** | 5.64 |
| | Survival | 9.52 | 9.47 | 9.09 | 9.13 | **8.78** |
| Group | Regression | 7.28 | 4.62 | **3.72** | 7.72 | 3.81 |
| | Binary class. | 10.69 | **4.71** | 5.23 | 7.95 | 6.30 |
| | Multiclass class. | 14.77 | 13.55 | **9.00** | 14.13 | 9.47 |
| | Survival | 188.61 | 27.14 | 24.86 | **9.14** | 24.71 |

Note:
For each combination of the predictor and outcome type, the median runtime for the analysis of one dataset is shown. Fastest methods are highlighted in bold.

method. On the Group data, Order (once), Order (split) and Partition performed about equally, Dummy and Ignore slightly worse. The runtime ranking was similar to the Digit data, except that Partition was faster.

The second row of Fig. 2 presents the simulation results for a binary classification outcome. On the Digit data, we observed a similar pattern of prediction accuracy as in the regression case. The Partition method was slowest to compute. Again, Order (once) was faster than Ignore. On the SNP data, all methods perform about equally. The differences in runtime results were also small, with Dummy being the fastest method. On the Group data, all methods except Ignore performed about equally. The runtimes were comparable to the Digit data.

The simulation results for a multiclass classification outcome are shown in the third row of Fig. 2. On the Digit data, the ranks of the methods were the same as in the regression case. However, the performance of the methods differed substantially. For the Digit data, the Ignore method misclassified a median of 83% of the observation, the Order (once) method only 3%. The runtimes and their differences increased, compared to regression and binary classification, but their order was the same. On the Group data, Partition, Order (split) and Order (once) performed about equally, Dummy and Ignore worse. The Order (once) method was considerably faster than the other methods.

The last row of Fig. 2 shows the results for a survival outcome. Here, the Dummy method performed best on the Digit and Group data and as well as the other methods on the SNP data. The other methods performed about equally. The runtimes and their differences increased considerably and the Dummy method was the fastest method on the Digit and Group data.

As expected, all results of Order (split) were not significantly different from those of Partition, except for the Digit data with a survival outcome. Order (once) performed

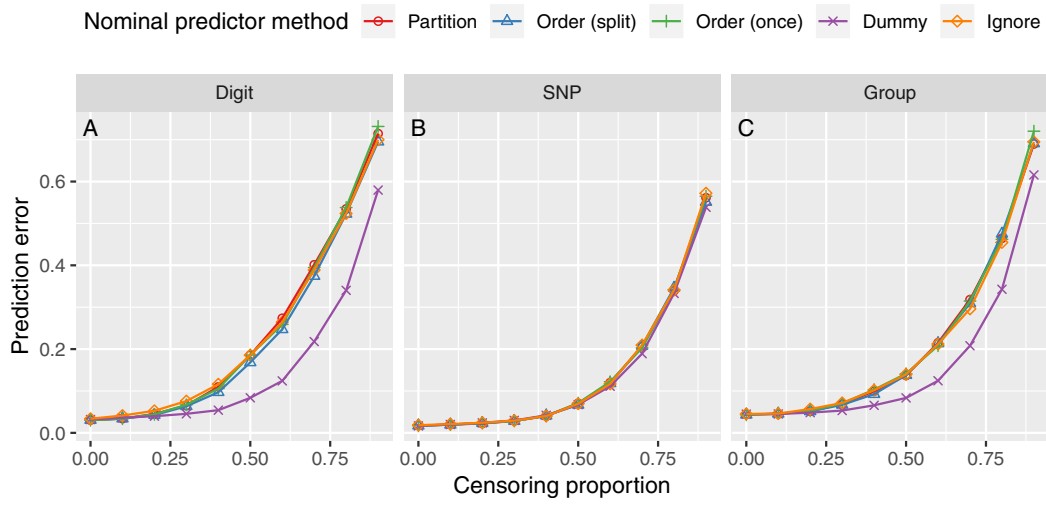

**Figure 3 Survival simulation results with varying censoring proportion.** The horizontal panels correspond to the predictor type (A, Digit; B, SNP; C, Group). Each line represents the prediction error for one method to handle nominal predictor variables. The prediction error is measured by the integrated Brier score.

significantly different from Partition on the Digit data with all outcome types, on the SNP data with continuous and multiclass outcomes and on the Group data with survival outcome. The results of Dummy were not significant on a binary classification outcome with SNP and Group data, but significant on all others. Ignore performed statistically different from Partition in all cases.

The results of the simulations with $n \in \{50, 200\}$ are shown in Figs. S1–S2 and Tables S2–S3. Compared to the results with $n = 100$, the ranks of prediction performance remained the same. The differences increased with $n = 200$ and decreased with $n = 50$.

The results of the simulation study with varying censoring proportions are shown in Fig. 3. As expected, the prediction error increased with higher censoring proportions. On the Digit and Group data, the Dummy approach performed best for higher censoring proportions. All other methods performed about equally. No appreciable difference was observed on the SNP data.

The results of the tree size and "absent level" simulations are shown in Figs. 4A and 4B, respectively. The left plot shows that trees grown with the Order (once) method are on average about half the size of trees grown with the Ignore method. This explains why Order (once) is computationally faster than Ignore in all multiclass simulations, despite the computational effort to order the categories of every variable. The right plot shows that the number of categories available in nodes drops fast with tree depth: For depth of 10 or larger, on average only two or three categories are available. With the Partition and Order (split) methods, no meaningful splits can be found for the absent categories, which reduces the prediction performance, as shown in Fig. 2.

## Real data analysis

Figure 5 and Table 4 show the results of the real data performance comparison. Boxplots of the runtime and results of statistical testing are shown in Figs. S4 and Table S4,
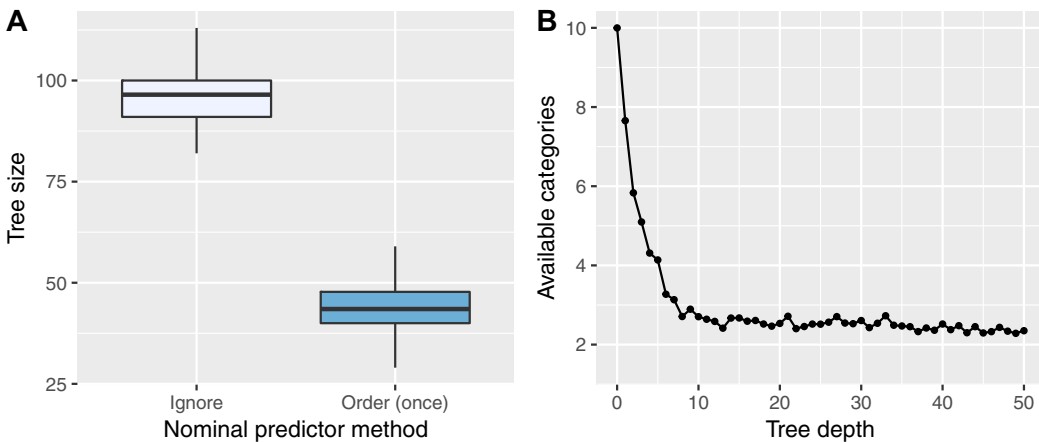

**Figure 4 Simulation results of the tree size (A) and "absent level" (B) simulations.** In (A), the number of nodes of 50 trees are shown for an RF grown with the Ignore and Order (once) methods. In (B), the number of available categories in nodes at different tree depths, averaged over 500 trees, is shown.

respectively. On the Tic-tac-toe dataset, the Partition, Order (split), Order (once), and Dummy methods performed about equally, the Ignore method worse. However, the dataset was easily separable, since all methods achieve low prediction errors. In terms of runtime, Order (once) and Dummy were fastest, followed by Partition and Ignore, while Order (split) was slowest. Comparable results were observed on the Splice data, except for Dummy, which performed slightly worse. The Order (once) method was fastest, followed by Partition, Dummy, and Ignore. Again, Order (split) was slowest. On the MPG data, no results could be obtained for the Partition method because the category limit was reached. The methods Order (split) and Order (once) performed best, followed by Dummy and Ignore. Here, Order (once) was about two times faster than Order (split) and Ignore and three times faster than Dummy. On the Servo data, the methods Partition, Order (split) and Order (once) performed about equally, Dummy and Ignore worse. The runtime differences were small, except for Order (split) which was slowest. A similar pattern was observed on the RA SNPs data: Partition, Order (split), and Order (once) methods performed about equally, Dummy and Ignore worse. On this dataset, Order (once), Ignore and Partition were fastest, Dummy and Order (split) slowest. On the AIDS data, all methods performed about equally in terms of prediction accuracy. Here, Dummy was the fastest method and Partition the slowest. Statistically significant differences from the Partition method after the correction of *Nadeau & Bengio (2003)* were observed for the Ignore method on the Splice, MPG, and Servo datasets as well as for the Dummy method on the Servo dataset.

## DISCUSSION

We compared five approaches to handle nominal predictors in RFs. The approach of sorting the categories once before analysis of the dataset (Order (once)) performed best or on par with other approaches in all simulations and on all real data examples, except for survival outcomes. The approach of ignoring the unordered nature of these predictors

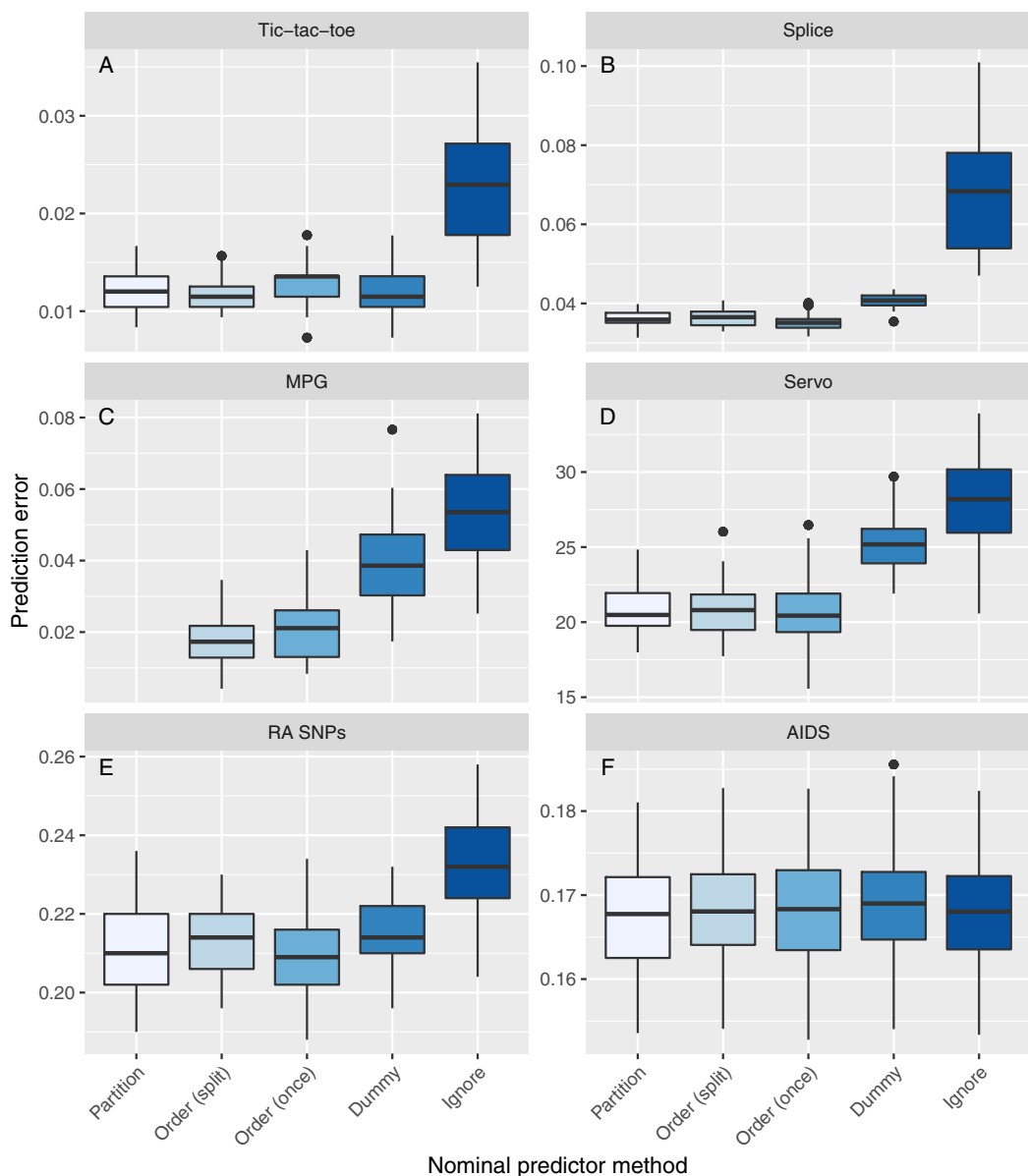

**Figure 5 Real data results.** The panels correspond to the datasets. Each boxplot represents the prediction error of one method to handle nominal predictor variables, applied to one dataset (A–F). The prediction error is measured with a 10/5-fold nested cross validation (repeated 50 times) by the mean squared error for regression and by the proportion of misclassifications for classification. No results could be obtained for the Partition method on the MPG dataset because the category limit was reached.

performed worst in all settings. The differences were generally large in regression and multiclass classification and smaller in binary classification and survival. On most datasets, the Order (once) and Dummy methods were computationally fastest. The new methods for multiclass classification and survival improved the prediction accuracy considerably.

Notably, Order (once) performed considerably better than Order (split) and Partition on the simulated Digit data with a multiclass outcome. This might be due to the fact that

| Table 4 Median runtimes of real data analyses (in seconds). | | | | | |
|---|---|---|---|---|---|
| **Dataset** | **Nominal predictor method** | | | | |
| | **Partition** | **Order (split)** | **Order (once)** | **Dummy** | **Ignore** |
| Tic-tac-toe | 19.46 | 26.82 | **18.65** | 19.37 | 21.60 |
| Splice | 105.40 | 241.44 | **90.62** | 107.71 | 112.48 |
| MPG | NA* | 8.89 | **5.14** | 17.02 | 10.55 |
| Servo | 6.56 | 9.70 | **5.38** | 5.98 | 5.51 |
| RA SNPs | 188.54 | 270.06 | **170.26** | 285.17 | 185.60 |
| AIDS | 67.61 | 49.64 | 47.58 | **16.49** | 47.08 |

**Notes:**
For each combination of the outcome and predictor type, the median runtime for the analysis of one dataset is shown. Fastest methods are highlighted in bold.
* No results could be obtained for the Partition method on the MPG dataset because the category limit was reached.

for this task it is only relevant if a digit is even or odd and thus a latent order of the categories exists. This order still holds if the categories are not present in a node in the training data and meaningful splits are still possible. In this case, the latent ordering cannot be found by considering all 2-partitions because the assignment of the empty levels does not change the impurity measure (*Au, 2017*). The same problem applies to the approach which sorts the levels by the outcome in each split.

A related problem often faced in real data are new categories in prediction. For example, a person in the prediction data might come from a geographical region which was not present in the training data or a rare genetic variant might appear first in the prediction data. If the levels are entirely new, most packages show an error that there are new categories, which were not available in the training data. It is intuitively clear that no meaningful splitting decision can be made for observations with new categories at a node splitting at this variable, regardless of the nominal predictor method. However, prediction should still be possible if there are other variables available. One simple approach would be to assign all observations with new categories to one node, another would be to assign them randomly. One might argue that the former is the more sensible approach because these observations are kept together and can be split by another variable afterward. However, *Au (2017)* showed that random assignment is generally more robust. Finally, if options to handle missing predictor values are available, new categories could be set to missing and handled appropriately.

Further research is required on how to handle nominal predictors in random survival forests. We proposed to order by log-rank scores (*Hothorn & Lausen, 2003*) once or in each split separately and achieved results on par with the partitioning approach. However, in two out of three simulation scenarios this was outperformed by dummy coding. No difference could be observed in the third simulation scenario and in real data.

Our results show that even for SNP data with only three categories, the splitting algorithm for unordered categories plays an important role. The simulation results indicate that dummy coding might be an alternative for SNP data. However, the results from the real data analysis show that the increase of the number of variables due to the dummy coding leads to reduced prediction performance and longer runtimes, and consequently

the ordering method is preferred. The real data results also show that ignoring the unordered nature of the predictors harms the predictions accuracy on real data, that presumably consists of SNPs with a variety of genetic models or no effect on the outcome at all.

## CONCLUSIONS

We have shown that ordering predictor categories a priori is a fast approach to handle nominal predictors in RFs which performs well on different kinds of datasets. In comparison to the standard approach, runtimes can be reduced substantially. Furthermore, in the case of a latent ordering, the prediction performance can be improved. We recommend to use this approach as the default when analyzing data with RFs, except for survival outcomes, where we recommend dummy coding.

## ACKNOWLEDGEMENTS

The authors are grateful to Ronja Foraita for valuable discussions.

### Funding

This work is based on data that was gathered with the support of grants from the National Institutes of Health (NO1-AR-2-2263 and RO1-AR-44422), and the National Arthritis Foundation. The funders had no role in study design, data collection and analysis, decision to publish, or preparation of the manuscript.

### Grant Disclosures

The following grant information was disclosed by the authors:
National Institutes of Health: NO1-AR-2-2263 and RO1-AR-44422.
National Arthritis Foundation.

### Competing Interests

The authors declare that they have no competing interests.

### Author Contributions

- Marvin N. Wright conceived and designed the experiments, performed the experiments, analyzed the data, contributed reagents/materials/analysis tools, prepared figures and/or tables, authored or reviewed drafts of the paper, approved the final draft.
- Inke R. König conceived and designed the experiments, authored or reviewed drafts of the paper, approved the final draft.

### Data Availability

 R scripts to replicate the simulation and real data study are available in the Supplemental File. Some raw data was provided to us by the National Arthritis

Foundation. We received permission to re-use the raw data for the purpose of this project, but not to share the original data.

## Supplemental Information

Supplemental information for this article can be found online at http://dx.doi.org/10.7717/peerj.6339#supplemental-information.

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
