# Peer review of "Splitting on categorical predictors in random forests"

_PeerJ, doi:10.7717/peerj.6339_

## Round 0.1 · original submission · Major Revisions

The three reviewers have all expressed some interest in your manuscript, but they have also raised a number of important points that need to be addressed. I'll ask you to go through all three reviewers' comments and respond to each point. I’ll summarise some of the key points below, but this is not to suggest that their other points do not also require addressing.

Reviewer 1 has noted that more detail about the ordering algorithm could be presented and I think that this would be helpful to many readers.

Their suggestion around presenting more information for the runtimes is one I strong support.

Perhaps the major change needed is around the scope of the simulations. Reviewers 1, 2 and 3 have all raised the possibility of expanding the simulations in terms of parameters (specifically sample size for Reviewers 2 and 3 and censoring for Reviewer 1, but the frequency of events for binary and multiclass outcomes could also be a candidate). Reviewer 3 notes that the number of scenarios and real data sets for the multiclass and survival outcomes could be increased, with the last point also made by Reviewer 1. In order for the recommendation of the manuscript to be accepted by (bio)statisticians and data scientists, showing that the results are robust is necessary. At the moment, I'd have to consider the findings to be interesting but not persuasive.

There are also some interesting questions from Reviewer 3 around performance times and it would be useful to see how these, as well as prediction error, vary between the approaches with different sets of parameters.

Reviewers 1 and 2 also ask about whether or not the prediction errors are statistically significantly different across the approaches and this should be simple to incorporate in anticipation of this question from at least some readers.

Reviewer 1 ·

Basic reporting

The English language should be improved. For example: “the exact same splits” line19, 54-55, 109; “of for” 52; line 61 “these exact methods”; difficult to understand lines 52-55.

In the Introduction the authors describe the variety of applications of random forest and the problem which arises while nominal predictors are in data. The description is too detailed and its large part is then repeated in Methods section. There is no literature review concerning the methods (heuristics) proposed to solve the problem with nominal predictors - the list (without any description) of relevant references is given in Methods section (lines 118-119).

Experimental design

The paper aims at investigation whether a new stage in the random forest induction procedure, concerning the ordering of nominal attributes, can make the algorithm faster with the prediction performance not worse than the standard approach.

The following points need to be addressed:
1.The description of the method of Coppersmith et al. (1999) should be more detailed. Points 1-3 (lines 122-129) are very general. Since the described method is essential in the paper, it would be worth to describe the way of calculating the probability and covariate matrix more precisely.

2.Line137: is “.. we identify the largest quartile…” – is it true? Maybe it should be written “the smallest”.
In case of survival data: What happens if one of the categories will have only censored cases?

Simulation studies:
3. Could you please describe more precisely the way survival data are simulated. I mean the sentence: “ For survival outcomes, the sum of …. a standard normally distributed variable was used as survival time with random censoring” (lines 184-186).

4. The censoring rate was set to 30%. It would be interesting, to investigate how the performance of the applied algorithm depends on the percentage of censored observations.

5. For survival data, the integrated Brier score was used as a performance measure. From the simulation description I can conclude that test datasets also contain censored observations. In experiments based on simulation data we do not need to use censored test data, since the true survival time can be calculated for all the observations. So even the model is trained using censored data, the performance measure should be calculated for uncensored test data. This does not apply to real survival data, in which the true survival is unknown.
Please, recalculate the performance measure.

Validity of the findings

1.In Table 3 and 4 beside the median runtime, also the min and max or upper and lower quartile should be presented. Please bold the best results.

2.Real datasets were narrowed to classification and regression problems. Since the conclusions concern also survival data, the real survival datasets should also be taken into account.

4.Statistical analysis of the results was not performed. The statement from the Discussion section “The new method for multiclass classification and survival improved the prediction accuracy considerably” was not proved for survival data.

Additional comments

The idea of ordering nominal attributes seems to be interesting and can cause the random forest algorithm to be faster, but the results presented in the paper should be extended with new datasets, especially survival ones. This type of data, due to the presence of censored observations, should be analyzed more carefully (see the points above).

·

Basic reporting

This paper discussed the splitting of categorical predictors problem in random forest. This paper has shown that ordering of categories before splitting is subtle but vital in random forest modeling, particularly in multiclass classification and survival analysis.

Generally, this paper is well written and easy to follow. The research is sound and the simulation and real data results are clearly presented.

Experimental design

Minor comments:
In "Simulation studies"
Line 198: "For each dataset, we simulated 50 training observations and 50 validation observations." The sample size for simulated training data might be somewhat small. It might be suitable for regression and binary classification. However, it might not be appropriate for multiclass classification and survival analysis. I suggest to set sampsize of training data to at least 100.

Validity of the findings

Minor comments:

Tables in Tables 3 and 4, and Box plots in Figures 2 and 3 are useful, but might not be the most appropriate method for multiple comparison across different methods. I would suggest to use multiple comparison statistical tests to check whether there are significant differences among these methods.

Reviewer 3 ·

Basic reporting

No comment

Experimental design

'Rigorous investigation performed to a high technical & ethical standard.'

In the simulation scenarios, the sample size for the training and validation data sets was small, especially for multiclass outcomes (50 observations in each sets for a total of 100 observations – see Table1 and line 198). Why this single choice of a small sample size to evaluate the performance of the proposed method? How this choice could explain some of the results?

There were also very few scenarios with multiclass and survival outcomes (only two each). There were also two real data sets with a multiclass outcome and none with a survival outcome. The authors should consider doing more simulations with different scenarios to evaluate the performance of their proposed methods.

Why the ‘ignore’ method is not always the method with the lowest runtime and ‘Partition’ with the largest runtime in all cases considered in the manuscript, as expected?

In practice, we expect the ‘Partition’ method to be the method with the best performance. However, according to the results of the simulations (Figure 2) and of the real data sets (Figure 3), it is not always the case. Why? A more in depth analysis of the results should be done to explain these results.

Validity of the findings

'Conclusion are well stated, linked to original research question & limited to supporting results.'

Lines 70-71: ‘Compared to the standard approach, the performance can be improved while simultaneously reducing computational time.’ There is a need to better explain how the performance can be improved compared to the standard approach. As mentioned above, in general it should not be the case; the chosen simulation scenarios might be too specific.

Lines 282-283: the sentence ‘The new methods for multiclass classification and survival improved the prediction accuracy considerably.’ overstates the results presented in the manuscript.

Additional comments

Other minor comments:

Line 64: ‘between a category and the outcome’ should be ‘between a categorical variable and the outcome’?

Line 122: what is the meaning of ‘N’ for the k x c contingency table here?

---

## Round 0.2 · Minor Revisions

Thank you for your revisions, which have addressed the reviewers’ and my questions well, and thank you to the reviewers for re-reviewing the manuscript.

There is just one small point raised by one of the three reviewers about the level of significance used in hypothesis testing (“two-sided p<0.05” based on supplementary table captions, but as the reviewer suggests, this is always worth mentioning explicitly in the methods). This could be a simple addition around Line 214, e.g. “These tests were performed at the two-sided 0.05 level.” Or “Two-sided p<0.05 was considered statistically significant.” This could be repeated or referenced at Line 261 when discussing the second set of tests. (I appreciate that Tables S1–S4 make this clear already.)

There are also a few minor suggestions/queries I have around wording in places and a request about some graph axis labels. Once the above and below are resolved, I will be very happy to recommend your manuscript be accepted.

Line 133: Should this read something like “number of observations who died or were censored before”. I’m not convinced that “died” works as an adjective in the original, but “dead” would be acceptable for a simple substitution as another option.

Line 170: I think “…it has been discussed…” would be better here.

Lines 182–183: This is uniformly distributed in the code and that point should be mentioned here.

Line 202: I think “…set to the sum…” would be better here.

Line 229: “datasets” (adding “s”)

Line 243: Shouldn’t this be “affliction” rather than “affection”?

Line 282: Just “methods” (plural and so no possessive apostrophe).

Line 298: You might like to consider “appreciable” instead of “considerable” here.

Figure S3: The y-axes for the top three panels are not very useful with only one value indicated. You could look at including 1 and 100, for example, for these to give some context to the scale.

Figure S4: Same issue with the top two panels on the left and the bottom two on the right. I’d add values to each of these as appropriate to provide at least three values on each axis (only one more would be needed for this for the bottom right panel).

Reviewer 1 ·

Basic reporting

The authors addressed all the questions. Only one problem is partially solved - statistical analysis of results. The authors have informed in the Real data analysis section about the statistical tests that were used, but without the value of significance level. It is important information. In the description of results, I haven’t noticed any sentence that contains the information whether the differences are statistically significant or not.

Experimental design

No comments.

Validity of the findings

Comments in the Basic reporting section.

Additional comments

No comments.

·

Basic reporting

no comment

Experimental design

no comment

Validity of the findings

no comment

Additional comments

The paper has been greatly improved after major revision. New data and comparison results make the work solid. I suggest its acceptance.

Reviewer 3 ·

Basic reporting

The revised manuscript meets the standards.

Experimental design

The revised manuscript meets the standards.

Validity of the findings

The revised manuscript meets the standards.

Additional comments

The revised manuscript is much improved.

---

## Round 0.3 · accepted · Accept

Thank you very much for your revisions which I think address all of my and the reviewer’s comments. I do have a few very minor copyediting comments about the edited/added text but these are so few and so minor that I’ll leave them for you to address in the proofing stage. I am delighted to accept your manuscript and congratulate you on writing what I’m sure will be a well-read article.

For the additions (Lines 215–216 and Line 263):

“Differences with p<0.05 were considered statistically significant.”

I would suggest that in the proof you revise this to:

“Differences with two-sided p<0.05 were considered statistically significant.”

simply to reassure the reader that you are using the (correct) two-sided tests and not the (inappropriate here) one-sided versions.

Related to this, for Line 295, I’d insert “two-sided” before the alpha, or delete “at alpha=0.05” altogether as you already mentioned this level of significance earlier. The same would then apply on Line 334.

I’d also suggest “outcomes” for the first instance of “outcome” on Line 298 (this refers to the continuous and multiclass outcomes).

Well done again!

#